# Biophilic Experience-Based Residential Hybrid Framework

**DOI:** 10.3390/ijerph19148512

**Published:** 2022-07-12

**Authors:** Eun-Ji Lee, Sung-Jun Park

**Affiliations:** 1Department of Architecture, Keimyung University, Daegu 42601, Korea; yej@stu.kmu.ac.kr; 2Department of Architectural Engineering, Keimyung University, Daegu 42601, Korea

**Keywords:** biophilic design, human nature connectedness, residential environment, hybridization, hybrid framework

## Abstract

There has been increasing academic interest in biophilic design in response to recent environmental and climate change issues, including the COVID-19 pandemic. However, discussions of the utilization of digital technology in providing universal access to nature, and opportunities to experience more diverse nature, are lacking. This study aimed to compare and analyze major theoretical systems for biophilic experiences in a residential environment, and to propose a hybrid framework that combines physical and digital design techniques for comparison and analysis. This paper discusses framework application strategies in line with scales of residential environments. Based on a systematic literature review, this study integrated and derived key elements of biophilic experience for a better quality of life in a modern residential environment and proposed a hybrid framework and strategy based on this. As a result, a hybrid framework of 15 integrated factors for three biophilic experiences was derived, and various strengths and potential opportunities were identified in terms of application depending on the scales. At the unit scale, it was found that the well-being and health of residents improved; at the building scale, the potential for sustainability was highlighted; at the complex scale, there was a contribution to higher residential competitiveness in multi-dimensional aspects. In particular, the biophilic experience-based hybrid framework in this study provided insights into addressing the weaknesses and threats discussed in the existing biophilic design.

## 1. Introduction

Recently, environmental impacts on human beings have become an important research topic, with the importance of nature being increasingly emphasized due to our interest in nature and aspirations for a healthy and fulfilled life. In particular, since the introduction to academia of the biophilia theory [1], which relates to the human instinct to seek connections with nature and all living things, there have been active attempts to conceptualize positive factors related to nature and living things. Kellert et al. [2] presented a biophilic design to connect humans and nature in the built environment, and discussed strategies to improve the health and well-being of urban dwellers, by providing opportunities to restore the relationship between humans and nature. The benefits of biophilic design are presented with a variety of empirical evidence, ranging from biological and mental health and well-being, to environmental sustainability and economic efficiency. Although the benefits of biophilic design have been emphasized as being more important to vulnerable or marginalized groups [3], and biophilic design has been applied in hospitals and facilities for children or the elderly, there is insufficient discussion of daily architectural spaces such as residential environments. However, as climate change issues (e.g., environmental pollution and drought, extreme cold or heat waves, and natural disasters) have intensified globally, there are wider threats to human health and welfare [4]. In addition, as the “untact” (un + contact) and “hometact” (home + contact) cultures have spread due to the prolonged COVID-19 pandemic, daily activities have been increasingly carried out inside the house; in addition, as physical distancing has been imposed outside the house, people have had fewer opportunities to experience nature. In other words, although the need and demand for biophilic design continue to increase due to climate change and COVID-19, it is difficult to present a practical alternative, as the environmental conditions and spatial scope for its application are more limited. As long as disease and climate change persist due to the diminished recovery function of human beings and the environment, the pandemic will become indigenous. In this sense, it is necessary to explore bio-friendly countermeasures from a multilateral perspective and to discuss the application of biophilic design, while focusing on the environmental crisis faced by humans and the corresponding universal welfare.

In our rapidly urbanizing modern society, it is difficult to effectively implement biophilic design [5], as there are increased limits due to spatial constraints of housing and geographical conditions of dense residential complexes. Nevertheless, one of the reasons for the lack of discussion around digital technology in terms of nature-based (biophilic) design stems from the idea that nature is in opposition to technology. Early biophilic design addressed the integration of the built environment and nature in the physical range, and emphasized the introduction of “authentic” nature with minimal intention and technical assistance [6]. However, as most design processes are mediated by digital information technologies such as building information modeling or 3D virtual modeling [7,8], there have been recent discussions on the application of immersive technology, such as Virtual Reality (VR) or Augmented Reality (AR) during the biophilic design process. Although immersive technology is still used as an experimental tool, by comparing the virtual biophilic environment with the non-biophilic environment, or real environment, [9,10], or by verifying independent effects through the simulation of biophilic factors [11,12], significant results have been achieved, indicating that the experience of nature through VR showed a similar level of positive response to reality [11,12,13]. Therefore, this study used a hybrid approach to combine the physical and digital planning elements of biophilic design, while focusing on the residential environment, by paying attention to the possibility of expanding the biophilic design experience and the need to improve accessibility. A hybrid approach can obtain characteristics beyond the original elements by mixing elements that have been recognized as heterogeneous in a complete form; that is, it encompasses the process of finding an appropriate selection and combination of mixed elements [14]. The goal of biophilic design is to improve Human Nature Connectedness (HNC); thereafter, a careful technical intervention is required based on the theoretical framework of biophilic design. Therefore, it is important to identify key biophilic design elements in the residential environment and to find an appropriate combination, while considering the physical and digital expression characteristics and the scale of the residential environment. In this study, the hybrid concept is considered for converting the housing technology, which is more focused on energy saving, into productive resources for residents, while alleviating the physical limitations of biophilic design.

This study aimed to compare and analyze key theoretical systems for biophilic experience in a residential environment and to propose a hybrid framework that integrates physical and digital planning techniques for comparison and analysis. To achieve the purpose of this study, we addressed the following Research Questions (RQ):

RQ1: How are the conceptual meaning and theoretical system of biophilic design defined and systematized/specified?

RQ2: How can biophilic design experiences contribute to residential environments and the quality of life (QoL) of residents?

RQ3: How can the residential hybrid framework that combines physical and digital planning be constructed, and what is its strategy?

This study attempted to clarify the conceptual meaning of the theoretical system and criteria of biophilic design, and to propose an integrated approach based on biophilic experiences to identify more important factors in the modern residential environment. Furthermore, to expand the biophilic experience and alleviate the physical limitations of the biophilic design plan, this study proposed a hybrid framework combined with digital technology.

## 2. Methodology

### 2.1. Research Schematics

Figure 1 indicates the method and scope to achieve the goal of this study, and the details are presented in the following. 

First, the residential environment factors and nature-related theories were considered as they are related to a better quality of life. Furthermore, previous studies related to the biophilia theory and the concept of biophilic design based on the theory, the theoretical systems, and the hybridization of the residential environment and biophilic design were analyzed. Second, after selecting keywords to integrate the various approaches of the biophilic design, we identified relevant materials through databases such as Google Scholar, Scopus, and Web of Science. Third, in this paper, we discuss how the definition and theoretical system of the biophilic design concept have been developed through the analysis of previous studies, and propose an integrated framework to identify more important factors in the modern residential environment. Fourth, based on the biophilic experience-based integrated framework, the expression characteristics and criteria of physical and digital plans for the hybridization of residential environments are derived. Finally, the application strategies and potential benefits per scale (i.e., unit, building, and complex) of the residential environment are discussed.

### 2.2. Literature Review

This study considered various search, screening, and selection methods for publications related to biophilic design. Figure 2 illustrates the method and scope of the literature review, and the details are presented in the following.

In this study, we selected five keywords, namely, “biophilic design”, “biophilic experience”, “biophilic building”, “biophilic architecture”, and “biophilic community and city”, to analyze various approaches to biophilic design in the field of architectural planning (Figure 3). The keywords were searched for in databases such as Google Scholar, Scopus, IEEE Xplore, and Web of Science, and titles and abstracts were screened by targeting collected documents, excluding duplicates. Through this, additional searches were conducted for publications from the potentially related literature, and the criteria for final selection were as follows: (1) a study investigating the impact of biophilic design through empirical or experimental research methodologies; (2) a study that discusses biophilic design planning elements and application methods in the architectural field, while focusing on more than one physical and digital planning element; and (3) a study in which the presented results are applicable to the scale of residential environments.

### 2.3. Analysis and Synthesis

This study explores the integrated approach of biophilic design in the built environment and discusses strategies in which a hybrid approach combining physical and digital planning can contribute to the residential environment and the quality of life of residents. The main analysis and summary of the publications are outlined below.

First, we compared and analyzed the theoretical approaches and systems of biophilic design. The classification systems are closely related to the organization of knowledge; this is because a theory or concept is based on the process of being categorized and classified [15]. In this respect, a literature review helps to indicate the networks among relevant authors and the results of an academic exchange over time. Therefore, this study analyzed the biophilic design system and contents that were mainly adopted by researchers, and extracted key factors that were treated with importance according to the application scale of biophilic design.

Second, the approach and key elements of biophilic design for a better quality of life for residents were analyzed. This study derived the physical, psycho-social, and economic factors of the residential environment to improve the quality of life of residents and analyzed the biophilic design factors that contributed to it. Here, we discuss the value of the residential environment for the health and well-being of residents, derive the environmental factors that are closely related, and compare and analyze the various approaches and benefits of biophilic design. This was done to identify more important biophilic design elements in the modern residential environment and to understand their support areas and methods.

Third, for the hybridization of biophilic experiences in the residential environment, the expression characteristics and components of physical and digital plans were systematized. Although the existing discussion on biophilic design contributed to broadening our understanding of the scope of the physical plan, it was necessary to analyze various methodological attempts because the extended scope of applying biophilic design and its possibilities are currently being discussed. Therefore, this paper discusses the biophilic experience from an integrative perspective and through a hybrid framework that combines physical and digital planning of a residential environment.

## 3. Related Works

### 3.1. Nature and the Residential Environment

Throughout the history of architecture, nature was seen as an uncontrollable environment that has a confrontational relationship with human beings, who must protect themselves from climate change risks and surrounding dangerous animals. In the residential environment, nature was a necessary reason and motivator, in addition to an object that had to be overcome. This is closely related to the survival of mankind and to the evolutionary process, and is consistent with the hypotheses that humans can be protected from external threats, or that they prefer natural landscapes that are favorable for survival [16,17]. As the field of architecture began to explore the relationship between humans and the environment, and human preferences and responses to nature, a well-documented range of corresponding benefits for humans was discovered. Therefore, this study considered the theories that explain the relationship between humans and nature, as shown in Table 1.

The theories explaining the relationship between humans and the natural environment can be divided into two perspectives. The first theory is about the preference for nature from a perspective of habitats and dwellings. The human mind, body structure, and capacity are formed through the process of living in the natural world, and as a result of evolution over a long period, they respond positively to environmental conditions favorable to threats and for survival. From a biological perspective, the theory based on the recovery of nature emphasizes human biological and emotional responses to nature. Humans are biologically and emotionally much healthier when exposed to nature than in an urban environment, and their function is optimized as they continue to experience it. That is, nature can have a positive effect on human health and well-being through a lifestyle that is integrated with nature and corresponding instinctive responses, and the effects can be further strengthened through repeated experiences.

In addition to theoretical concepts in terms of preference and recovery, related studies and the literature suggest that contact with nature has a positive effect on cognitive, physiological, and physical health. Exposure to nature and natural analogs leads to positive emotions away from negative emotions and thoughts, and these changes have been demonstrated through responses such as blood pressure, heart rate, and electroencephalographic activity. In particular, involuntary attention to nature is related to stimulation of the five senses. Perception of the environment is not only visual but also involves various human senses such as touch, hearing, and smell, and the more that positive stimuli occur at the same time, the higher the level of recovery due to involuntary attention [21,26]. Diette et al. [27] investigated the effect of natural sounds on pain relief in surgical patients, and suggested that the older the patient, the lower the level of pain felt during or after surgery. In addition, visual stimuli related to natural scenery and living things are effective in reducing respiration and blood pressure [28], and have a positive effect on perception-related responses such as reduction in anxiety and tension, fatigue, and malaise [29,30]. The human brain functionally responds to sensory patterns and elements found in the natural environment. When interacting with nature, concentration, thinking, and creativity are improved, and factors related to memory decline are suppressed [31,32]. Such effects are similarly shown in VR-based virtual natural environments [33], and are mainly presented quantitatively through EEG responses and cognitive function tests [34,35]. In a previous study on physical function improvement, Detweiler et al. [36] observed and investigated the correlation between the frequency of outdoor garden use and fall accidents in dementia patients. The results showed that, the higher the frequency of garden use, the lower the number of falls. The severity of the falls was also reduced by approximately 30%. All natural environment experiences, including artificially created nature, are closely related to the improvement of walking function and balance [37], and measures to link them with related programs, such as physical training and rehabilitation, are actively discussed [38].

As a metaphorical tool for proposing the diversity of experiences and recommended “nature time” based on environmental scales, the “Nature Pyramid” [39] explains that the everyday neighborhood natural environment is crucial in determining a healthy lifestyle. Mother Nature, at an international and national scale, is located at the top of the pyramid and guarantees a high level of value and rewards but cannot be applied at all scales because equal access to nature cannot be guaranteed. Therefore, it is crucial to have appropriate “nature time”, based on the environmental scales in terms of access to nature in the built environment and the idea that the natural elements around humans should be considered first. In the Nature Pyramid, the residential environment provides various potential opportunities for the well-being and health of residents and is recognized as an important factor in determining quality of life. In a situation where people have fewer opportunities to experience nature due to social problems such as the recent pandemic, the residential environment can function as an opportunity to re-establish the connections with nature. In this study, we derived the residential environment factors related to a better quality of life, as shown in Table 2, to identify the potential of the residential environment as a mediating environment for HNC improvement, and to discuss the value of nature in the essential relationship between humans and the residential environment.

The concept of housing can be explained by the entire connections of spatial, temporal, and social and cultural experiences, along with spatiality. The residential environment is divided into the phenomenal environment, the behavioral and experiential environment, and the contextual environment, which are composed of the human environment and the physical environment [50]. In a broad sense, the residential environment can be defined as tangible and intangible external conditions, including economic, social, physical, and psychological conditions [51]. Therefore, in this study, we defined the residential environment as a concept that encompasses the economic and socio-psychological aspects of residents, in addition to the physical aspects. The indicators for measuring the quality of life of residents in relation to the residential environment are mainly discussed based on individual satisfaction. The quality of life can be understood as a subjective evaluation of the objective conditions and environment necessary to lead a fulfilled life [52], since it is composed of multidimensional aspects and diverse relevant factors that are presented based on the perspectives of researchers. In this sense, we identified a total of 19 elements, based on previous studies that investigated residential environmental factors related to the quality of life of residents. Among these, the factors having overlapped meanings were integrated and reconstructed, and finally, ten factors were derived. Detailed contents and categories of the derived factors (including integrated factors) were organized together as elements.

Physical factors are related to residential performance and include safety, convenience, accessibility, and comfortability. In particular, it has been reported that comfortability, such as safety against crime or accidents, indoor environmental quality (IEQ), and securing sunlight, have a significant impact on quality of life [43,45,46]. Relationships, sentience, security, and leisure factors were derived from a socio-psychological perspective. Many papers on relationships and leisure, encompassing factors such as cultural facilities and services, shared space, and intimacy with neighbors, discuss significant relationships with improved QoL of residents [47,48]. In terms of socio-psychology, related prior studies [53,54] focused on “bond of human and animal” to improve QoL, suggesting that animal-assisted activity is more effective than entertainment and other leisure activities. Accordingly, the socio-psychological space and programs of the residential environment should be further subdivided, and this needs more attention when creating a residential environment from a biophilic point of view. Finally, in the case of economic factors such as management/efficiency and added value, studies on subdivision of factors or dealing with specific elements are relatively scarce, but it appears that a significant relationship exists for the improvement in QoL [42,47]. In particular, the QoL index is subjectively evaluated by an individual’s ability or perception, and economic factors appear to have a strong influence on health, emotion, and overall life satisfaction.

### 3.2. Biophilia and Biophilic Design

The social psychologist Fromm (1973) [55] mentioned “biophilia” along with the concept of love for nature, and it was later popularized through the biophilia hypothesis of the biologist Edward O. Wilson (1984) [1]. Biophilia is a combination of “bio-”, meaning life, and “-philia” meaning love or kindness. It has the innate tendency in which human beings who have evolved from nature place value on all life and all living organisms. That is, the biophilia hypothesis seeks to restore the relationship between nature and humans based on the emotional partnership of humans inherent in natural life [18]. This means that human beings have a biological desire to connect with nature at the physical, mental, and social levels, and when the inner biophilia is activated, personal well-being and productivity are improved. Exploring life and feeling a bond with its biological processes are associated with a positive human response.

It is important to truly understand the value of biophilia through a fundamental background in neurology and biology. The human brain is composed of the sympathetic nervous system and the parasympathetic nervous system, and the natural balance of sympathetic and parasympathetic nerves is the ideal state. However, in a chaotic and unstable environment, the sympathetic nerve induces a way of thinking, such as in the case of a struggle or escape, and the parasympathetic nerve is suppressed, causing mental fatigue, such as in the form of stress and frustration. Conversely, when exposed to the natural environment, the sympathetic nerve is suppressed and the parasympathetic nerve is activated, leading to an ideal state, and the improvement in physical function and concentration, and psychological stability. Previous studies [23,56,57,58] investigating experiences with nature or exposure to nature in this regard showed that the natural environment and natural factors contributed to strengthening human immunity and physiological functions, and that habitats with less exposure to nature were more likely to result in fatigue and disease. That is, the core of the biophilia hypothesis is that humans have evolved so that they do not need to be sensitive or attentive to stimuli that are innately familiar, such as in the natural environment. As such, human beings have developed their senses and tendencies to match the environmental characteristics of nature and their way of life; as a result, in just a few decades, as the size of the urban population around the world has rapidly increased, the construction process has highlighted efficiency and convenience, and mankind has moved away from nature [29]. Similarly, advances in building technologies have also led to the separation of residents from nature to reduce harmful environmental impacts, with an emphasis on smart buildings rather than on the occupants [59]. That is, problems have occurred that often overlook the intrinsic value or spatial significance of the residential environment during the modernization of architecture. Accordingly, recent studies have been undertaken to integrate biophilic design with sustainable architecture, green buildings, and zero-energy buildings; however, there is still a dearth of studies that suggest specific application methods for the practice of these designs. 

Biophilic design is a strategy for interpreting biophilia and introducing it to the built environment, while focusing on the restoration of the relationship and bond between nature and humans [60]. Although the aforementioned approaches of sustainable or eco-friendly buildings focus on less-harmful environmental impacts and, essentially, the saving of energy, biophilic design differs in that it emphasizes human energy production through the enhancement of HNC [61]. Biophilic design was defined by the ecologist Stephen R. Kellert [62], and is a concept developed and analyzed by experts in biology, psychology, and architecture. The goal of biophilic design is to create a built environment as an optimal habitat for the health and well-being of human beings, and the design can be seen as an evidence-based physical environment plan and an applied science approach. From a biophilic design point of view, a built environment that lacks a sense of nature cannot provide optimal healthy conditions for humans and society, and a process that integrates nature is required for a more regenerative and resilient living environment. Resilience is the capacity of a system to absorb and utilize psychological fluctuations and changes to derive benefits, and has a structure that is continuously maintained [63]. Accordingly, the application scope of biophilic design has been extended, not only to the indoor environment, but also to the architectural structure, the exterior of the building, and the city, and appropriate application methods based on these scales have been discussed to establish a resilient urban environment.

### 3.3. Hybridization and Potentiality

As a concept, a “hybrid” relates to genetic hybridization derived from biology, and it means a combination or mixture of elements that cannot be combined; it also means a state in which each element is combined and one element cannot dominate the whole [64]. The dictionary meaning of the term “hybrid” has a negative expression, such as in the sense of genetic failure or a crossbreed hybrid [65]. Today, however, as the value of the difference between new and unfamiliar things has spread, it has been interpreted as the stage or process developed from the perspective of function and morphology. That is, a hybrid mixes heterogeneous elements in a complete form, while also creating new values or possibilities [14]. In modern society, “hybrid” is not a term used to describe specific styles or “-isms”, but has been used as a means of integrating digital technologies and information in the processes of globalization, localization, and multiculturalism.

Hybrids have emerged in slightly different forms in line with urbanization and changes in lifestyle, social value, and architectural technology in architecture. Hybridization in early architecture was accomplished by mixing culture and style according to globalization, and subsequently, hybrid architecture was mainly used to mean “for multi-purpose use” [66]. In the same context, hybrid architecture is formed naturally when utilized for purposes that differ from those of the initial plan, but it also occurs during classifying of new spaces by suppliers, when diverse demands from users increase [67,68]. In the field of building technology, hybridization refers to the building system that encompasses the combination or simultaneous use of processing units within a building to operate at the highest efficiency, while highlighting sustainability-based building efficiency and performance [69]. As the physical and digital factors of architecture have combined due to digital informatization, and the boundaries and domains of space have become blurred, the hybrid concept in a broader sense has been discussed. In particular, the hybridization of physical–digital factors of spaces has accelerated rapidly over the past two years (in a short period), starting with COVID-19, as most of our living activities, such as education, work, and leisure, have become concentrated in the residential space. When temporal and spatial boundaries become blurred due to hybridization in a residential environment, the role of a hybrid medium that connects them is particularly important. This is because a hybrid medium can provide better spatial meaning and a satisfying experience to residents; it must also be able to satisfy expectations of humanity and sustainability in terms of the future residential environment. This study emphasized the possibility of biophilic experience as a new medium for a hybrid residential environment.

As the efficient development of space using digital technology is being promoted, related studies are discussing biophilic experiences using immersive technologies, robotics, and Natural Language Understanding (NLU) [5,33,70,71]. Particularly, in the field of environmental psychology, VR and AR are being used to effectively control experimental environments and variables, and to quantify human responses to them. Immersive technologies such as VR and AR enable the simulation of virtual natural environments and elements, the investigation of biological responses according to conditions [9,12,72], and a comparison with the real environment [73,74]; as a result these technologies have resulted in new approaches to biophilic design research. Other studies using NLU, which have contributed to the extended scope of biophilic experiences, also deal with direct and indirect interactions with virtual natural elements through facial expressions, behaviors, and psychological states [70,75], and investigate emotional and cognitive responses of social care robots similar to real companion animals [76,77]. The results of this study proved that an experience with nature mediated by digital technology and virtual elements also had a profound effect on human health and quality of life, suggesting the possibility of hybridization of biophilic experiences.

## 4. Biophilic Experiences for Residential Environments

### 4.1. Theoretical Frameworks of Biophilic Design

The theoretical process of biophilic design started with the conceptualization of “nature” in architecture so as to practice biophilia, and was introduced through eight architectural characteristics [78]. Since then, Kellert [62] defined biophilic design through two dimensions, six elements, and 72 attributes of biophilic design; this is known as an interpretation system to help understand the biophilic design. Furthermore, Cramer and Browning [79] suggested three preliminary categories for a practical approach to practicing biophilic design; Terrapin Bright Green, a construction consulting company, specified these with 14 patterns [80], which were recently increased to 15 patterns [81]. The biophilic design system should be modified and developed gradually in line with the uses and users of architecture [6]; to this end, Kellert [6] suggested three types of biophilic experiences and 25 factors, along with the aforementioned biophilic design system. Regarding the practical cases for the evaluation and certification of biophilic designs, certification programs exist, such as the Living Building Challenge (LBC) and the WELL Building Institute’s Standard. The WELL [82] certification system is the first certification system that considers human health and comfort, and deals with the biophilia theory in the “Mind” concept. In obtaining WELL certification, biophilia is used as a concept for better mental health of users, and although it does not cover many items, it is an important factor contributing to the meaning and originality of projects. The LBC [83] is a certification system developed by the International Living Future Institute (ILFI); six factors and 72 attributes of biophilic design suggested by Kellert are actively introduced in this system, which requires a high level of performance in the built environment and throughout the living environment.

In this study, four main theoretical frameworks [6,62,81,83] were selected, as they were mentioned the most frequently in the literature reviews related to the interpretation of biophilic design. Based on this, the perspectives and approaches of each author on the interpretation of biophilic design were the focus, and the categories and detailed elements were compared and analyzed. Figure 4 shows the top categories and sub-factors of the four theoretical frameworks.

Since the elements and attributes initially classified in Kellert’s interpretation [6,62] include relationships with nature, attitudes, and perceptions, they can be understood as conceptual characteristics for biophilic design theory. This is a detailed specification that explains the biophilic design, encompassing all the comprehensive content that can be used to understand nature in the built environment. Therefore, there is an ambiguous side to practicing this for architectural practitioners, and a possibility in which the very subdivided hierarchical structure will lead to a rather limited scope of application guidelines. In this sense, Kellert integrated the following three biophilic experience types, and suggested 25 elements that are concise and easy to understand: (1) direct experience of nature; (2) indirect experience of nature; and (3) experience of space and place. The biophilic design experience is a set of options for a better relationship between nature and humans in the built environment, and construction practitioners or users are encouraged to utilize it concretely and appropriately.

The LBC is a certification program that was developed in 2006 and is a tool applicable to all scales of projects, ranging from new and remodeled buildings, to landscapes, cities, and society [84]. The LBC certification guideline is continuously updated and consists of seven petals and 20 imperatives based on LBC 4.0 [83]. The LBC fully adopts the biophilic design theory, but it can contribute to a wider understanding of the hybrid framework of this study because it includes environmental impact and performance based on sustainability and related techniques. Furthermore, by providing project participants with specific examples and guidelines for integrating the built environment and biophilic design, various approaches are suggested to respond to problems occurring in the project process.

Browning and Ryan [81] developed a framework based on both scientific evidence and feasibility to bring nature into the architectural space, and finally presented 15 patterns. They scientifically verified the physiological and mental benefits of biophilic design in the fields of environmental psychology and neuroarchitecture, and addressed natural concepts to introduce the results into the built environment. Biophilic design patterns are divided into the following based on the characteristics of nature and spatial perspectives: (1) nature in the space; (2) natural analogues; and (3) nature of the space. Key architectural design elements are identified as examples of applications for each pattern.

### 4.2. Supportability of Biophilic Design for Residential Environments

This study examined the supportability of biophilic design for residential environment factors related to the quality of life, functional application, and effective practice of biophilic experience in a residential environment. This study analyzed the support provided for the residential environment factors, which were contemplated through the literature review, by the empirical evidence for the biophilic design practice plan and its advantages. In addition, based on the understanding of the key framework and interpretation of biophilic design, the most relevant detailed elements per factor were derived.

As biophilic design is a strategic tool for HNC, it is worth noting that there is a difference in the support level for residential environment factors including multidimensional characteristics. Therefore, in this study, three (*, **, ***) support levels were classified based on the direct and indirect benefits and influences of biophilic design for ten residential environment factors. Table 3 indicates the residential environment factors and the supportability of biophilic designs that contribute to them.

As a result of the analysis, various benefits related to a better quality of life for residents were found when the biophilic design was applied to the residential environment. Among a total of ten residential environment factors, four of these (i.e., accessibility, comfortability, sentience, and management/efficiency) showed strong supportability, and the other three (i.e., safety, relationships, and added value) were found to have a direct significant contribution through biophilic design. The remaining three factors (i.e., convenience, security, and leisure) are partially or indirectly supported through biophilic design. Considering the development of smart home technology and the diversity of immersive content, it is judged that more opportunities will potentially be available in the future.

In the process of deriving the results, as shown in Table 3, we identified key factors related to residential environment supportability among the factors presented in the four main frameworks of biophilic design. In addition, although the authors adopted different terms, the four proposed frameworks contain elements that are similar to, or overlap with, each other, and it was judged that the biophilic design experience and attributes presented by Kellert [6] are relatively representative of each framework. This is thought to provide a clearer criterion for defining the ambiguous concept of the natural environment because the three categories of experiences focus on the subject who experiences it.

### 4.3. Biophilic Experience-Based Integrated Framework

This study focused on the categorical classification concept of biophilic design experience suggested by Kellert [6], and performed curation work to systematize the key framework for the residential environment. Curation is a method for selecting more necessary and important information as the selection area expands amid the overload of information and roles, and refers to any process of adding value through selecting, refining, arranging, etc. [101]. In this study, the overlap and similarities between the four frameworks were compared and analyzed from the perspective of biological taxonomy [102], and reduction, refinement, simplification, and categorization were performed. To this end, we identified the key elements of the four frameworks contributing to residential environmental factors related to QoL, and the final framework was derived through consultation and review by five experts in architecture and interior design based on the curation results. Figure 5 shows the biophilic experience-based integrated framework and derivation process.

In this study, 15 integrated elements for biophilic experiences in a residential environment were derived, and each factor was modified and supplemented with terms encompassing four key framework categories. The biophilic experience-based integrated framework of this study includes key elements of biophilic design for a residential environment, and deals with simplified and categorized integrated categories and elements. Direct experience of nature is about experiencing the characteristics and features of the natural environment in multiple senses, and is similar to the “Nature in the Space” category in the 15 patterns of biophilic design [81]. This study integrated elements with similar characteristics by focusing on the contact characteristics of each element. For instance, “air” and “thermal and airflow variability” have similar contact and inflow methods in the built environment and can be experienced simultaneously. Regarding “weather”, it can be connected with various elements including “water” and “light”, but it is important to recognize external “weather” as being separated from the built environment and to perceive the changes in the weather [6,62]; in this study, classification was made based on visual features such as view.

Indirect experience of nature is about establishing associations with nature and subjective feelings in a metaphorical manner. Expressions that have been continuously used through architectural styles or “-isms” were subdivided, and repetition and reiteration between each element were commonly found. We particularly considered the class hierarchy and meaning of terms, to encompass detailed factors such as “shell and spirals”, “arches”, and “domes”, and complex factors such as “age, change, and the patina of time”, and “information richness”. We also distinguished the features of the functional imitation of nature (i.e., biomimicry) and simulation (e.g., artificial lighting and HVAC).

Experience of space and place is about experiencing the spatial and locational characteristics of nature that contribute to human biological evolution. This study focused on accessibility to residential spaces, and the consciousness of residents regarding residence and placeness. Specifically, considering that the elements related to the experience of space and place in the main framework are difficult to interpret and practice [13,103], it is necessary to discuss the digital planning technique in this study. Accordingly, in this study, biophilic content and experiential programs in the virtual environment are also included in the experience of space and place. In particular, although the biophilia and education elements in this study are covered by one framework (i.e., the LBC standard), biophilia is not a single instinct, but a repetitive learning rule for cultivating and functioning [1]; as continuous learning and experience are important, it is necessary to address it seriously. 

## 5. Residential Hybrid Framework for Biophilic Experience and Strategies

### 5.1. Biophilic Experience-Based Residential Hybrid Framework

In this study, a literature review was conducted and previous research cases were analyzed from an integrative point of view. By focusing on physical and digital expressions of biophilic experiences in a residential environment, as shown in Table 4, a residential hybrid framework based on biophilic experiences is proposed. 

In the biophilic experience-based residential hybrid framework of this study, the features of physical expressions include architectural planning and spatial factors; in the case of digital expressions, since they are linked with home services and systems, related sensors and devices are included. A hybrid dwelling based on biophilic experience can provide residents with a new biophilic experience while complementing the restraints of modern cities and dwellings on the premise of a mixture of physical and digital expression techniques. This framework can be understood as a network configuration for practicing this hybrid biophilic experience, and the expressive features in line with each biophilic experience element are closely connected with each other. That is, it provides a residential environment in which one can be immersed in the relationship with nature and its experiences, by emphasizing the biophilic physical structure and space, the appropriate arrangement of detailed items, and supporting or realistically simulating it.

A hybrid residential environment for biophilic experience requires knowledge of detailed fields such as landscape and public design, architectural structure and system, and interior design and home service. However, there have been insufficient attempts to link biophilic design in these fields, and accordingly, in the field of biophilic design (see Figure 3), more attention must be paid.

### 5.2. Strategies by Residential Scale: Strengths and Opportunities

An important task in the practice of biophilic experiences in residential environments is to provide appropriate combinations of biophilic properties optimized for residents depending on scales, among a wide range of biophilic factors having various benefits. This study divided the residential environment into three scales (i.e., unit, building, and complex) in order to identify strategic planning methods that should be seriously considered depending on the application scale. Unit represents the actual living space of a household, and includes a living room, bedroom, and kitchen; building represents a multi-family house or apartment, and includes the shared spaces of each household (e.g., lobbies), moving paths (e.g., corridors or stairs), and rooftop gardens, building exterior, or system design; finally, complex represents the communal facilities and main entrances for nearby residents, and pedestrian and road plans within a complex.

This study analyzed the positive effects of three scales in terms of “Strengths and Opportunities” in order to derive application strategies based on the physical and digital expression characteristics of the biophilic experience-based hybrid framework, and to suggest applicability and evidence for practice. In this study, a “Strength” refers to an initiative that can be effectively implemented when applying the biophilic experience-based hybrid framework; an “Opportunity” is defined as a resource to improve latent possibilities of development or vulnerabilities. Table 5 shows the strategies of a residential hybrid framework for biophilic experience and strengths and opportunities based on three scales. 

An experience with nature has different effects and influences depending on the scale and may appear differently depending on the individual’s sociodemographic and cultural background [116,117]. However, there is a lack of previous studies discussing the linkage and differentiation methods of biophilic design according to scales in specific building environment settings, and there is a tendency to overlook the technical benefits of biophilic design. The biophilic experience-based residential hybrid framework in this study suggests the possibility of supplementing some of the weaknesses [80,116,117] discussed in existing biophilic design. In addition, it can help prevent negative perceptions of nature described in the HNC-related theories. “Biophobia” [115] refers to the defensive attitudes and fears that can occur in certain natural elements, and is common in some people when they experience living creatures such as snakes, spiders, and pests, or disaster threats such as storms and droughts. Biophobia, like biophilia, has different levels and impacts per individual; therefore, the digital technology in this study can provide a customized experience to residents, so one can be immersed in a positive biophilic experience according to one’s personal choice and preference.

Specifically, when applied according to the scale of the residential environment, each unit can directly contribute to better access to nature and fewer physical restrictions (e.g., maintenance/manageability and geographical conditions) and help provide more opportunities for improved health and wellbeing of residents. At the building scale, there are supports for a win–win relationship between humans and nature while improving overall building performance and obtaining natural resources. Regarding the complex scale, clues can be provided to solve the problems of land availability and supply, which have triggered concerns in the existing biophilic design discussion; these are contributions to securing residential competitiveness and increasing property value. Such strengths and opportunities should be discussed in more detail in related technologies and detailed fields in the future, and sufficient quantitative and qualitative research needs to be undertaken based on the results of this study.

## 6. Discussion

This study proposed a biophilic experience-based residential hybrid framework and expression characteristics as a strategy for all urban dwellers to access nature daily. To this end, the existing literature on biophilic design, which is discussed at various scales, was reviewed and analyzed. As a result, we presented a strategy of a residential hybrid framework for biophilic experience and provided the analysis of strengths and opportunities according to the scale of the residential environment. The results of this study are summarized as follows.

Numerous research papers [77,105,123,124] that were additionally identified during the literature review address the broad range of evidence supporting biophilic design; however, these studies were not identified as studies of biophilic design. In particular, evidence regarding technical benefits for biophilic experiences requires an understanding of more complex design types for biophilic design and a careful analysis of additional direct and indirect effects. Therefore, this study clarified more important factors in the residential environment through the four main frameworks [6,62,81,83] of biophilic design.

Biophilic design showed a difference in the support levels according to ten residential environment factors for a better quality of life for residents. Regarding indoor environmental quality (IEQ), it was identified that there were supports for pleasant and comfortable environments, and diverse potential benefits to manage the supports and ensure they function effectively. Another important factor is the sensibility towards nature; numerous papers [5,120,125] were found to discuss the application of biophilic design for sensory deterioration due to aging and the development of the growth period. Furthermore, although supports for such factors as convenience and leisure are weak, theoretical and technical supports for related benefits are required, considering the importance of the indirect benefits of biophilic design [35].

In the process of analyzing the key elements of biophilic design for a residential environment, this study focused on the experience of biophilic design suggested by Kellert [6], identified key factors within an integrated framework for biophilic experiences in a residential environment, and modified and reconstructed terms. For example, in the case of “water” [6,62], “responsible water use” [83], and “presence of water” [81], the overlapping meanings between existing framework elements were synthesized; when applying “geomorphology and fractals” [62] and “non-rhythmic sensory stimuli” [81] in practice, we focused on minimizing ambiguous interpretation. Based on three types of biophilic experiences, we derived a total of 15 framework elements, focused on the hybrid concepts needed to effectively practice them in a modern residential environment, and analyzed physical and digital expression characteristics based on the literature review.

The biophilic experience-based residential hybrid framework of this study was divided into physical and digital design techniques, and is intended to provide new value and differentiated experiences for existing biophilic design. The physical and digital expression features of the framework were partially interdependent, and when using them in combination, greater synergy can be obtained. This study provides a strategic analysis according to the scales of the residential environment to apply the proposed framework. Regarding the analysis, when applying the biophilic experience-based residential hybrid framework, we identified the following as opportunities for improvement: strengths per scale; biophobia [115], which was discussed as a negative effect in existing biophilic design; boredom due to fixed shapes and repetitive patterns [80]; and subjective differences in biophilic design effects [116]. Specifically, at the unit scale, the framework was found to be able to directly contribute to the improvement in occupants’ health and well-being; at the building scale, the sustainability of higher building performance and economic benefits were emphasized; and, at the complex scale, there were contributions to common interests, such as greater land availability and enhanced competitiveness of residential areas.

## 7. Conclusions

This study compared and analyzed the conceptual meaning of biophilic design and major theoretical systems for biophilic experiences in a residential environment. Biophilic design began with the conceptualization of nature in architecture, and has been systematized in terms of various interests, such as architectural and spatial expression characteristics, subjects who use these characteristics, or designers who practice biophilic design. However, the theoretical system for biophilic design has been continuously revised and developed. It is integrated into a broader meaning that encompassed users’ emotions and experiential characteristics beyond the limited concept of architecture, and seeks new values and perspectives on the relationship with nature in modern society. Accordingly, this study identified the key elements of biophilic design from an integrated perspective to realize the new value of biophilic experience in a residential environment, and analyzed the supportability of biophilic design for 10 residential environment factors related to resident QoL. In particular, by comparing the various advantages of practicing biophilic design according to residential environment factors, it was shown that biophilic design can directly or indirectly contribute to the improvement in quality of life. In the sense that the link between existing biophilic designs and digital technology provides direct benefits to the overall satisfaction of residential life, such as accessibility, comfortability, and relationships, in the field of smart homes and immersive content, an active interest and development strategies for biophilic design are required.

The relevant literature and previous studies discussed the possibility of linking digital technologies and physical support for biophilic experiences, but there have been insufficient attempts to convert the potential and related benefits into a clear design framework. However, since immersive technologies are utilized as an experimental means in empirical research on biophilic design and human responses, there is potential for further research related to this study.

The biophilic experience-based integrated framework of this study contributes to reducing the gaps that may arise in the process of interpreting and applying key biophilic design frameworks and providing residents with a more effective biophilic experience. Furthermore, the attempt to explore hybrid strategies to improve biophilic quality and expand the biophilic experience of residents in future residential projects is valuable as it represents a new attempt in the related field.

The residential hybrid framework of this study emphasizes the existing physical structure, space, and proper arrangement of detailed items, and presents the digital sensors and devices required to automate or simulate it. The hybrid framework proposed in this study is a mixing process for physical and digital design, which have been recognized as heterogeneous. It is a proposal for appropriate selection and combination, and it should be recognized that it can be continuously developed and updated along with related technologies and additional supporting data.

This study discusses the strengths and opportunities that may be apparent when applying the residential hybrid framework to biophilic experiences by subdividing the scales of the residential environment. This can provide insights that enable the identification of physical and digital strategies, and more important hybrid components, that should be considered for the hybridization of biophilic experiences at diverse scales. Based on the results of this study, future research should compare and analyze residents’ responses according to specific application rates of physical and digital techniques, and more interdisciplinary knowledge is required to link hybrid strategies and resident benefits. In particular, it is necessary to share weaknesses and threats in the relevant technical field, and the practice of biophilic design, and to discuss methods to address them.

## Figures and Tables

**Figure 1 ijerph-19-08512-f001:**
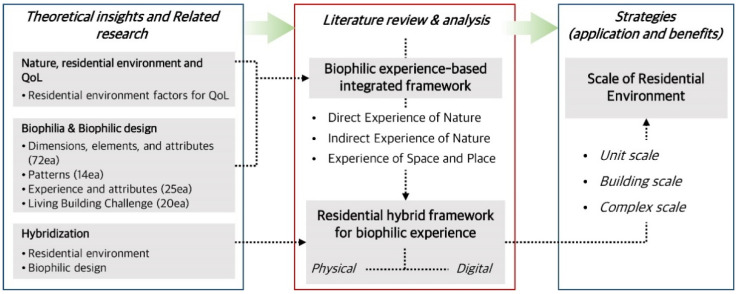
Research scheme.

**Figure 2 ijerph-19-08512-f002:**
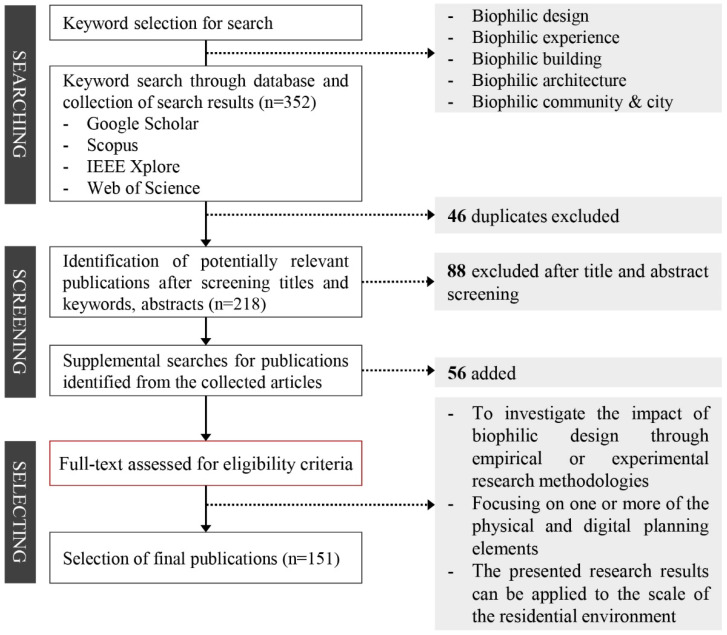
Method and scope of the literature review.

**Figure 3 ijerph-19-08512-f003:**
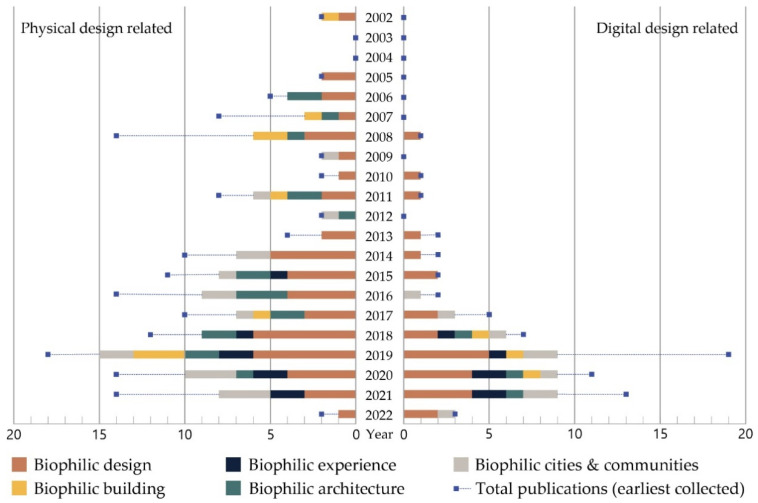
Selected publications and related keywords.

**Figure 4 ijerph-19-08512-f004:**
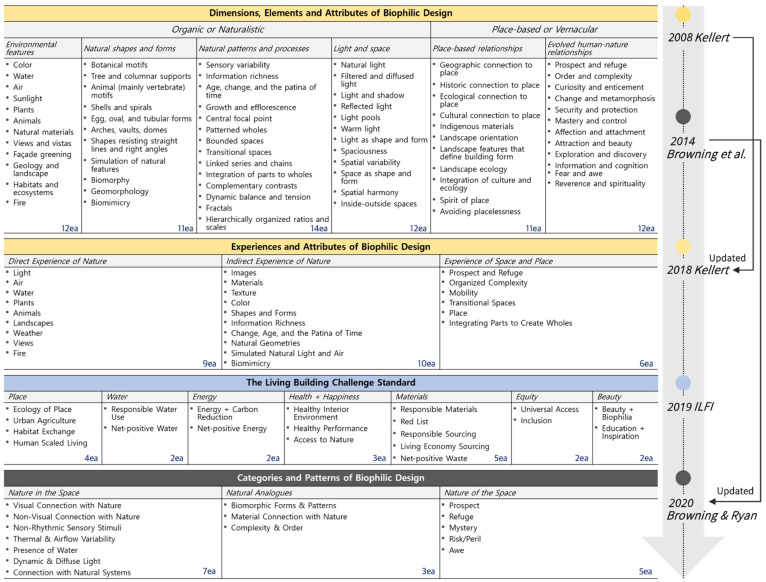
Main theoretical frameworks of biophilic design [6,62,80,81,83].

**Figure 5 ijerph-19-08512-f005:**
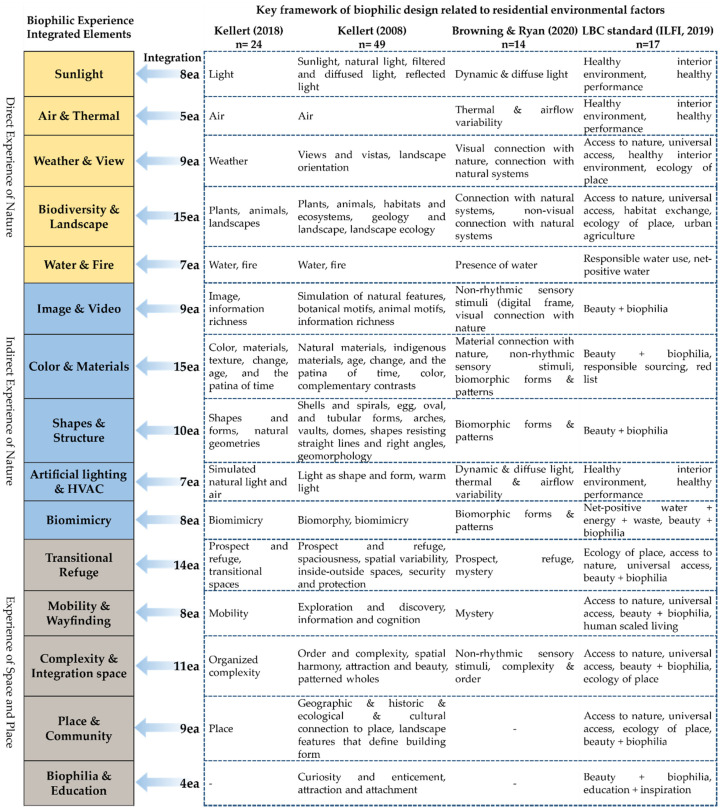
Biophilic experience-based integrated framework and process [6,62,81,83].

**Table 1 ijerph-19-08512-t001:** Theories regarding the relationship between humans and the natural environment.

Perspective	Theory	Description	Resource
Preference for nature from a perspective of habitat and dwelling	Savanna Theory	The nature of the African rainforest (savanna) is the most preferred landscape genetically as the landscape element in which mankind was born.	[18,19]
	Evolution Theory (The Aesthetics of Survival)	As mankind evolves, landscape elements favorable for survival can be identified and classified according to five characteristics (i.e., prospect, refuge, enticement, peril, and complex order).	[17]
	Prospect and Refuge Theory	Humans focus on the characteristics of the prospect to find and collect risk factors, and on places of refuge to protect themselves against external threats.	[16]
	Place Attachment Theory	While describing emotional connections to places, a “sense of place” and “sense of community” are suggested.	[20]
Recovery of nature from a perspective of human biological response	Stress Recovery Theory	Stress is restored from such elements as vegetation, water, and views as psychological and physiological responses to all nature and situations that do not threaten a healthy life.	[21,22]
	Attention Restoration Theory	Through a natural environment that is involuntary attention, one can recover from the fatigue caused by directed attention.	[23,24]
	Understanding and exploring	A higher level of preference and recovery can be provided through the process of understanding (i.e., coherence and legibility) and exploration (i.e., complexity and mystery) of the natural environment.	[25]

**Table 2 ijerph-19-08512-t002:** Residential environmental factors related to improving QoL.

Factor		Elements	Resource									
			[40]	[41]	[42]	[43]	[44]	[45]	[46]	[47]	[48]	[49]
Physical	Safety	Safety against crime/gas leaks/fall and fracture/vulnerable environments	●	●	●	●	●	●	●	●	●	●
	Convenience	Convenience of movement and furniture arrangement/parking facilities/public convenience services/smart home services	●	●	●			●	●	●	●	●
	Accessibility	Access to medical, welfare, and transportation facilities/neighboring natural environments and green parks	●	●	●	●		●	●	●	●	●
	Comfortability	Ventilation/sunlight/lighting/noise/temperature/humidity/hygiene comfort	●	●	●	●	●	●	●	●	●	●
Socio-psychological	Relationships	Relationships with and proximity to neighbors/attachment to a place of residence/pet environment	●	●		●	●	●	●	●	●	●
	Sentience	Clarity of visual/audible information/tactile and olfactory satisfaction	●	●		●	●		●	●		
	Security	Home security level/securing privacy and private spaces		●	●	●	●		●			
	Leisure	Cultural and welfare programs/communal gardens/resident exchange and activity spaces	●	●	●	●	●		●	●	●	●
Economic	Management/efficiency	Ease of residential maintenance and management/energy saving and production (renewable energy)/flexible spaces and multifunctional furniture			●	●	●	●	●	●	●	●
	Added value	Increase in real estate and property value/improved regional competitiveness		●	●				●	●		

Note: Factor (integrated factor). Safety (safety). Convenience (public service, public infrastructure). Accessibility (nearby nature, location). Comfortability (hygiene, indoor environment quality). Relationship (attachment, intimacy). Sentience (aesthetic, perception). Security (security, privacy protection). Leisure (entertainment, leisure). Management/efficiency (management, efficiency). Added value (land availability, real estate value).

**Table 3 ijerph-19-08512-t003:** Residential environment factors and supportability of biophilic design.

Factors		Support Areas and Examples	Benefits	Resource	Most Relevant Elements
Physical	** Safety	- City parks and trails - Visualization of fire uses - Edible landscaping - Eco-friendly materials, and green walls	- Improvement of walking function and sense of balance - Lower crime rates and violence - Improved visual satisfaction - Improved food safety - Less respiratory problems and headaches	[6,30,37,85,86,87]	Geology and landscape, views, plants, fire, urban agriculture, material, visual and non-visual connection with nature, red list, etc.
	* Convenience	- Automation of temperature/humidity, lighting, view, etc., according to resident status	- Improved subjective satisfaction and stability - Optimizing thermal comfort	[30,70,88]	Light, water, weather, views, simulation of natural features, thermal and air flow variability, healthy interior, human scaled living, etc.
	*** Accessibility	- Physical and digital plans based on all biophilic designs	- Higher access to nature and higher satisfaction - Physical and mental healing - Higher life expectancy	[6,81,89,90]	Views and vistas, landscape, façade greening, ecology of place, access to nature, connection with natural systems, mobility, prospect, etc.
	*** Comfortability	- Location, shape, and area of the opening - Indoor gardens and landscaping - Greening walls and water spaces - City parks and greening	- Improved life cycle and biorhythm - Optimizing thermal comfort - Psychological recovery - Reduced urban heat islands	[11,57,91]	Landscape ecology, sunlight, water, weather, plants, air, views, healthy performance, dynamic and diffuse light, thermal and airflow variability, connection with natural systems, etc.
Socio-psychological	** Relationship	- Environments for companion animals and living creatures - Balcony or patios, and courtyards	- Emotional recovery - Less loneliness - Improved conversation and favorability	[76,77]	Inside-outside spaces, water, plants, animals, landscape ecology, habitat exchange, transitional spaces, beauty + biophilia, etc.
	*** Sentience	- Images, videos, and sounds of nature - Natural patterns, materials, and shapes - Blue spaces (flowing water)	- Less anxiety, tension, and frustration - Higher creativity and concentration	[32,90,92]	Sensory variability, texture, materials, shapes, and forms, information richness, non-rhythmic sensory stimuli, complexity and order, etc.
	* Security	- Windows that can switch transparency - Spaces that can switch to an open/closed status - A small rest space with a view	- Psychological recovery - Less stress - Improved subjective satisfaction and stability	[70,81,93]	Spatial variability, prospect and refuge, universal access, mystery, transitional spaces
	* Leisure	- Gardening activities - Community gardens	- More frequent use of public spaces - More exchange and conversation with residents - Securing food resources	[94,95,96]	Cultural connection to place, education + inspiration, urban agriculture, plants, animals, mobility, etc.
Economic	*** Management/efficiency	- Automated temperature/humidity, ventilation, and treatment systems - Collection of renewables and biogas energy	- Lower energy consumption and building maintenance costs - Less water pollution - Higher building ratings	[35,97,98]	Simulated natural light and air, light, net-positive water; energy; waste, biomimicry, etc.
	** Added value	- Green spaces near the residence - Street trees and vertical gardens	- Less air pollution and better air quality - Securing biodiversity - Higher real estate value	[35,99,100]	Landscape features that define building form, place, integrating parts to create wholes, awe, etc.

* weak supportive, ** moderate supportive, *** strong supportive.

**Table 4 ijerph-19-08512-t004:** Biophilic experience-based residential hybrid framework.

BEIEs		Residential Hybrid Framework			Resource
		Physical Expression Features	Digital Expression Features		
			Sensors	Devices	
Direct Experience of Nature	Sunlight	Glass walls; stained glass; light well; louvers	Vital signs; outdoor/indoor monitoring	Louvers/curtain controller (cont.); heliostat mirror ^1^	[60,80]
	Air and Thermal	Insulator; double skin; vents; ventilation windows; awning	Vital signs; outdoor/indoor monitoring	Window/vent cont.	[79]
	Weather and View	Geographical conditions of site; opening size/location; atrium; pavilion; courtyard	Outdoor monitoring	Smart window; stereo	[5,79]
	Biodiversity and Landscape	Flowerpot; green wall; green roof; aquarium; roof garden	Outdoor monitoring; multimodal	Social pet robot; smart aquarium/plant growers	[6,77,104]
	Water and Fire	Waterway; fountain; waterfall; wall fountain; fireplace; candle	Outdoor/indoor monitoring	Immersive screen ^2^; stereo; automatic watering;	[5,60,83,105]
Indirect Experience of Nature	Image and Video	Picture frame; painting; mural	Multimodal	Immersive screen; smart window/frame; stereo;	[6,75,106]
	Color and Materials	Inside-outside color; wood; stone; cotton; leather; natural; material	Multimodal; outdoor/indoor monitoring	Wall display; lighting cont.; media facade	[80,81]
	Shapes and Structure	Shells; oval; arches; domes; geomorphology; egg	Multimodal; outdoor/indoor monitoring	Wall display; media/kinetic facade	[60,70]
	Artificial lighting and HVAC	Light as shape and form; warm light; HVAC	Vital signs; outdoor/indoor monitoring	Projector lighting; smart window/lighting; automatic HVAC	[6,104,107]
	Biomimicry	Biomimicry; biomorphy	Outdoor/indoor monitoring	Natural process devices; bio energy systems; responsive building systems	[6,104,107]
Experience of Space and Place	Transitional refuge	Space in space; booth/tiered seating; parasol shades; balcony; curtain wall	Outdoor/indoor monitoring	Smart window/lighting; wall display; immersive screen; kinetic façade	[70,80]
	Mobility and Wayfinding	Corridor; void; stairs; access; ramp; sign	Multimodal; outdoor/indoor monitoring	Wall display; immersive screen; escalator; elevator; projection mapping systems; stereo	[105]
	Complexity and Integration space	Complex/integration of direct and indirect elements of nature	Multimodal; indoor monitoring	Immersive screen; projection mapping systems; HMD; EGD; stereo	[6,105]
	Place and Community	Local landscapes; neighborhood green links; spaces for communication of residents	Outdoor/indoor monitoring	Solar panel; rainwater recycling systems; digital façade	[80,83,104]
	Biophilia and Education	Education/experience space; signs	Multimodal; vital signs	Immersive screen; HMD; EGD	[83,108]

BEIEs = biophilic experience integrated elements, HVAC = heating, ventilating, and air conditioning, HMD = head-mounted display, EGD = eye glasses-type display. ^1^ Turns to keep reflecting sunlight toward a predetermined target, compensating for the sun’s apparent motions in the sky. ^2^ Display showing three-dimensional augmented media using light interference effects (hologram, invisible display) or non-material projections (fog screen).

**Table 5 ijerph-19-08512-t005:** Strategies of a residential hybrid framework for biophilic experience by scale.

BEIEs	Strategies for a Hybrid Framework		Strength and Opportunities of Application by Scale		
			Unit	Building	Complex
Sunlight	P	Various opening shapes/sizes/locations and spatial structures; appropriate building height for sunlight	- Improve natural lighting in highly dense urban areas and solve shading issues [6] - The combination of heliostat mirrors ^1^ and interior texture/pattern provides various environment settings for light and rich design sources	- Improve heat retention rates; energy saving - A biophilic façade where related facilities and devices can be integrated with the surrounding landscape of the building	- Expansion of residential complex using shading site and creation of resident support facilities [109] - Increase in land availability
	D	The inflow of reflected light in line with the amount/direction of sunlight; automatic louvers/curtain			
Air and Thermal	P	Vents for natural ventilation/purification, structure/arrangement; sunshade/louver to prevent overheating	- Solve the problem of contaminated air circulation through natural ventilation [110] - Improvement in occupant comfort/reliability considering both indoor and outdoor quality	- Improve overall building performance in terms of air quality/thermal comfort - Improve the aesthetics of the building when linked with the kinetic façade design	- High density of tall trees/plants with air purification functions contributes to air quality and the formation of urban “wind corridors” [111]
	D	Automatic ventilation and IEQ alarm depending on air flow/direction; modular or mobile air purification/shaded plant systems			
Weather and View	P	Openings to see nature outside and creating an environment responsive to weather changes	- Actual sound/image of the real-time surrounding environment, weather, and habitat information of living organisms enhances residents’ attachment to places [112] - The daily acquisition of natural information contributes to the improvement of the essential relationship with nature	- Providing real-time sky images, rain sounds, etc. in the lobby and main entrance enables easy and cost-effective immersive experiences - Customize/automate visual/audible media according to the season	- Public outdoor spaces that can recognize the external environment and weather within the complex contribute to promoting physical activities and extending lifespan [89]
	D	Visual/auditory stimulation of weather or external environmental conditions and information			
Biodiversity and Landscape	P	Landscape design for animal/plant habitats or biodiversity and reproduction	- Automatic management of the biological environment and the link with smart technologies reduce maintenance time and cost for residents. - An animal-shaped social robot supports social care for the elderly and disabled residents [77]	- Smart/automatic green walls improve air purification performance by approximately 50% [113] - Attached substrate-based system alleviates the structural defect problem of BD [114]	- Sequential landscaping such as complex landscaping and vertical gardens in the building enhance biodiversity - Parklet ^2^ [104] induces residents’ awareness and active participation in nature
	D	Social robots and creating automatic growth/management environments			
Water and Fire	P	Interior/exterior structure and decoration design using water and fire	- Diversify the experience of water indoors, such as the sea and waterfalls - Visual/auditory stimulation against the use of fire improves safety and protection	- Combination of vertical garden and outdoor landscaping with a rainwater system contributes to securing more water resources	- Provide diversity/variability/dynamics of water characteristics such as fountains and waterways - Creation of networks for excellent quality management
	D	Provide visual/auditory stimulation for the dynamic properties and usability of water and fire			
Image and Video	P	Visual decoration for realistic and metaphorical expression of nature	- A combination of physical space elements and displays reduces the initial cost of the biophilic experience - Customized virtual nature can mitigate biophobia ^3^ [115]	- Improvement of diversity/efficiency in providing nature-related images and video media	- Enhance the overall biophilic image within the residential complex
	D	Provide virtual nature through remote display windows/walls			
Color and Materials	P	Natural colors and coloration; Finishing design that reflects the characteristics and textures of natural materials	- Promote the variability of the fixed elements of space - In terms of cognitive aspects, it is possible to select colors and materials customized for residents considering individual differences [116,117] in the effect of biophilic design and to create databases. - Relieve the boredom [80] of repeating fixed shapes and patterns	- Utilization of common spaces and creation of various atmospheres according to demand - Enhance the availability of common spaces for residents according to changes in use and function	- Enhance the overall biophilic image in the complex - Improve the experience of finding ways/lighting landscape - Improve the quality of parks and public design within the complex
	D	Lighting or display that can control color/texture			
Shapes and Structure	P	Natural geometric shapes and forms, or pattern design			
	D	Digital facade for form elements such as walls, floors, and ceilings			
Artificial lighting and HVAC	P	Natural light spectrum and HVAC in artificial lighting	- Artificial lighting that reproduces real sunlight/moonlight spectrum enables the appreciation of the flow of time regardless of direction and structure - Regulation of resident biorhythm; improved sleep quality	- Combination of artificial lighting and automatic irrigation systems supports vertical and urban agriculture - Revitalize the local economy; secure safe food resources	- Improvement in lighting landscape in the residential complex
	D	Providing virtual light and shadow elements; automated HVAC based on occupant conditions			
Biomimicry	P	Morphological/material/functional solutions of architecture imitating biological characteristics	- Possibility of self-sustaining residential energy generation and natural treatment [107] using microbial culture and household waste - Reduction in living expenses	- Materials and building systems that mimic the biological advantages of organisms contribute to temperature/humidity/air quality control and water resource/biogas energy generation [118,119]	- More application of biomimicry technologies contributes to biophilic communities and sustainable urban environments
	D	Technical solutions that mimic biological properties			
Transitional refuge	P	Relaxing and hiding areas with views; structure and arrangement to connect indoor–outdoor environments	- Provides a space that satisfies both views and hiding through remote opening and closing and transparent/opaque switching windows	- Contribute to building aesthetics and energy efficiency when linked with kinetic façade/ambiance wall	- Creation of spaces for resident exchange such as open spaces and pavilions according to the opening and closing of buildings
	D	Automatic opening and closing and space change device, or system			
Mobility and Wayfinding	P	A sense of openness in moving spaces, such as corridors and stairs, and walkways; sign design using natural elements; creation of corridors for wild animals	- Virtual interaction elements according to movement and gestures enhance the sense of nature and resident experience	- Improvement of mobility and openness between buildings and exchange between residents through linking with public streets	- Corridors for wild animals and permeable sidewalks promote transportation convenience and biodiversity
	D	Virtual and responsive nature elements based on moving paths and location of occupants			
Complexity and Integration space	P	Complex and integrated design considering biophilic properties per space and layer (i.e., color, pattern, material, etc.)	- Providing visual/hearing/synesthesia of the natural environment using projection mapping alleviates temporal and spatial limitations such as virtual travel and memory recall - VR/AR content can be used	- Combination of materials, colors, patterns, etc. depending on the building contributes to the clues for finding ways and creating landmarks in residential areas - Improve the aesthetics of the building	- Multisensory complex parks such as ponds, green spaces, and lighting contribute to positive emotion induction and child development [86,120]
	D	Creating a complex environment in which one can be immersed in nature, by using spatial mapping, sound, and 3D holograms			
Place and Community	P	Architectural design based on the characteristics of the local community and the creation of a space for mutual exchange	- The use of local indigenous materials and the formation of intergenerational communities for recycling (online) contribute significantly to the improvement in residents’ environmental awareness [121,122]	- Indigenous plant-based modular roof landscaping and common vegetable gardens; automatic/customized management system - Preservation of indigenous ecosystems and securing safe food resources	- Secure resources according to the climatic and ecological conditions of the dwelling (e.g., artificial wetland and solar panel); landscape design based on local characteristics - Secure regional competitiveness
	D	Utilization of eco-friendly technologies and energy systems according to climate and geographic characteristics			
Biophilia and Education	P	Educational space and signs design for biophilia effects and architectural application characteristics	- Recognition of information about biophilic effects (energy, biodiversity, etc.) that occurred over a period of time - Cultivation/reinforcement of biophilia through repetitive learning patterns	- A shared space where one can experience the effects of biophilia (i.e., pulse, blood pressure, etc.) - VR/AR content can be used	- Acquisition of knowledge or information about biomimicry-based public design - Cultivation/reinforcement of biophilia through repetitive learning patterns
	D	Cultural and ecological experience environments and environment plans for experiencing biophilia effects			

BEIEs = biophilic experience integrated elements; P = physical; D = digital; IEQ = indoor environmental quality; HVAC = heating, ventilating, and air conditioning; ^1^ Turns so as to keep reflecting sunlight toward a predetermined target, compensating for the sun’s apparent motions in the sky; ^2^ Green spaces and pocket parks using parking spaces in creative ways for each residential complex; ^3^ Innate genetic tendency to develop fear or strong negative avoidance responses to certain natural stimuli, settings.

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
