# Peer review of "Biophilic Experience-Based Residential Hybrid Framework"

_ijerph, 2022, doi:10.3390/ijerph19148512_

Round 1
Reviewer 1 Report
This study reviewed and analyzed the theoretical systems for biophilic experiences in a residential environment, and propose a hybrid framework for biophilic design with physical and digital techniques. The hybrid framework with key factors for residential biophilic experience at unit, building, and complex scale was derived. This paper was well written and can make contributions to current research. I recommend a minor revision.
1. In section 3.1, I suppose review theories toward the relation between humans and nature were not enough. In my opinion, there are many literatures elaborated nature contact with physical activity (i.e., perspective of habitat and dwelling), and nature contact with human health such as chronic disease and body function (i.e., perspective of human biological response) which are not belong to stress, attention, or understanding. The theories reviewed may be not enough, and the author should cautiously summarize them.
2. In line 231, it is appreciated to provide the 19 residential environmental factors with QOL and how they were summarized as 10 factors.
3.In section 4.3, what are the principles you formulate the 15 elements based on previous four frameworks, what is the difference? Has it been verified by experts?
Author Response
Point 1: This study reviewed and analyzed the theoretical systems for biophilic experiences in a residential environment, and propose a hybrid framework for biophilic design with physical and digital techniques. The hybrid framework with key factors for residential biophilic experience at unit, building, and complex scale was derived. This paper was well written and can make contributions to current research. I recommend a minor revision.
Response 1: We are grateful for the careful review of this paper. We appreciate your opinion and have tried to reflect this.
Point 2: In section 3.1, I suppose review theories toward the relation between humans and nature were not enough. In my opinion, there are many literatures elaborated nature contact with physical activity (i.e., perspective of habitat and dwelling), and nature contact with human health such as chronic disease and body function (i.e., perspective of human biological response) which are not belong to stress, attention, or understanding. The theories reviewed may be not enough, and the author should cautiously summarize them.
Response 2: We reviewed additional previous studies on the physiological, cognitive, and physical benefits of contact and exposure to nature, and through this, section 3.1 were revised and supplemented.
Point 3: In line 231, it is appreciated to provide the 19 residential environmental factors with QOL and how they were summarized as 10 factors.
Response 3: In response to your comment, we have presented 19 residential environmental factors in Table 2. In addition, we have supplemented the derivation method and QoL.
Point 4: In section 4.3, what are the principles you formulate the 15 elements based on previous four frameworks, what is the difference? Has it been verified by experts?
Response 4: This study compared and analyzed the overlaps and differences between the four frameworks using the curation method from the viewpoint of biological taxonomy. After that, in the process of integrating into 15 elements, the final framework was derived through consultation and review of experts in the architecture and interior design fields. In response to your suggestion, we supplemented the methods and principles in Section 4.3, and Figure 5 was revised to depict the key elements of the four frameworks and 15 categories. We believe that this revision clarifies our approach and the scope of the analysis.
Reviewer 2 Report
The paper deals with the use of the Biophilic approach in residential buildings.
The Biophilic approach is a very current and interesting topic. The manuscript is well organized and the framework is described in a complete way.
The only suggestion are related to minor aspect:
-The paper is classified as article, however I think that it is more a review, so I suggest to change the paper type.
-The conclusions should be imporved and they should summarize the answers to the three research questions (RQ1, RQ2, RQ3) presented in the introduction section.
- In the manuscript there are some typos.
Author Response
Point 1: The paper deals with the use of the Biophilic approach in residential buildings. The Biophilic approach is a very current and interesting topic. The manuscript is well organized and the framework is described in a complete way. The only suggestion are related to minor aspect.
Response 1: We value your opinion and are grateful for your positive review of this paper.
Point 2: The paper is classified as article, however I think that it is more a review, so I suggest to change the paper type.
Response 2: As per your recommendation, we have considered changing the review type. However, this study focuses on deriving a biophilic experience-based residential framework from a hybrid design perspective and presents strategies according to the scale of the residential environment, rather than reviewing trends and the latest progress of related studies. Therefore, we conclude that the article type was more appropriate.
Point 3: The conclusions should be improved and they should summarize the answers to the three research questions (RQ1, RQ2, RQ3) presented in the introduction section.
Response 3: We have reconstructed and supplemented the sentence and context in the Conclusions section to summarize the answers to the research questions, reflecting your opinion.
Point 4: In the manuscript there are some typos.
Response 4: We have checked and corrected the typographical errors.